# Longitudinal Plant Health Monitoring via High-Resolution Mass Spectrometry Screening Workflows: Application to a Fertilizer Mediated Tomato Growth Experiment

**DOI:** 10.3390/molecules28196771

**Published:** 2023-09-22

**Authors:** Anthi Panara, Evagelos Gikas, Anastasia Koupa, Nikolaos S. Thomaidis

**Affiliations:** Laboratory of Analytical Chemistry, Department of Chemistry, National and Kapodistrian University of Athens, Panepistimiopolis Zografou, 15771 Athens, Greece; panaranthi@chem.uoa.gr (A.P.); vgikas@chem.uoa.gr (E.G.); ankoupa@chem.uoa.gr (A.K.)

**Keywords:** onion-based fertilizer, tomato leaves, plant growth, HRMS, time series, chemometrics

## Abstract

Significant efforts have been spent in the modern era towards implementing environmentally friendly procedures like composting to mitigate the negative effects of intensive agricultural practices. In this context, a novel fertilizer was produced via the hydrolysis of an onion-derived compost, and has been previously comprehensively chemically characterized. In order to characterize its efficacy, the product was applied to tomato plants at five time points to monitor plant health and growth. Control samples were also used at each time point to eliminate confounding parameters due to the plant’s normal growth process. After harvesting, the plant leaves were extracted using aq. MeOH (70:30, *v*/*v*) and analyzed via UPLC-QToF-MS, using a C18 column in both ionization modes (±ESI). The data-independent (DIA/bbCID) acquisition mode was employed, and the data were analyzed by MS-DIAL. Statistical analysis, including multivariate and trend analysis for longitudinal monitoring, were employed to highlight the differentiated features among the controls and treated plants as well as the time-point sequence. Metabolites related to plant growth belonging to several chemical classes were identified, proving the efficacy of the fertilizer product. Furthermore, the efficiency of the analytical and statistical workflows utilized was demonstrated.

## 1. Introduction

The agri-food needs of contemporary society have led to an international explosion of intensive agricultural practices. However, the necessity for environmentally friendly methods to ensure agricultural biosafety and protect human health has become eminent. The implementation of fertilizers produced via composting offers an alternative method for increasing the agricultural production yield in a less invasive way, taking advantage of their beneficial effect on soil and plant nutrition [1] and contributing to the cyclic economy by recycling organic waste [2].

In the terms of the current study, an onion-based fertilizer manufactured via the hydrolysis of an onion compost product is evaluated. The fertilizer was applied to tomato plants at five time points, covering all anticipated developmental stages of the tomato plant’s life cycle, in order to investigate plant health and growth. This fertilizer was comprehensively chemically characterized in a previous study by our research team [3]. The selection of tomatoes, as a model organism, was driven by the plant’s financial importance, as well as its widespread adoption as a reference organism for omics analysis [4,5,6,7,8,9]. Tomato (*Solanum lycopersicum*) is a *Solanaceae* family member and an exceedingly well-known and financially significant crop globally [10]. Interestingly, the tomato market is expected to grow at a CAGR of 4.76% from USD 197.76 billion in 2023 to USD 249.53 billion by 2028 [11].

A wide range of analytical techniques have been used in the literature to identify a diverse range of compounds in tomato plants. High resolution (HR) techniques, such as nuclear magnetic resonance (NMR) [7,12,13] and mass spectrometry-based (MS-based) methods such as gas chromatography–HRMS (GC-HRMS) [5,14] or liquid chromatography–HRMS (LC-HRMS) [14,15,16,17] have been employed for the determination of metabolites in tomato plants.

In order to evaluate the efficacy of a fertilizer, the plant subject’s phenotypic characteristics should first be evaluated. Equally important is the assessment of the biochemical alterations that occur in the plants due to the application of fertilizer, which should be monitored via chemical analysis. The determination of specific molecular species is not as effective as holistic fingerprinting, with the latter providing access to the widest possible array of biochemical changes induced by the implementation of the novel soil compost improvement product. Therefore, metabolomics is employed in such cases [18,19], as this field has the potential to uncover the underlying biochemical processes in a hypothesis-free manner. Nevertheless, the majority of metabolomic workflows analyze a control group versus a treated one, or, at most, a limited number of treated groups. A small number of studies can be found in the literature concerning time-series metabolomics. The main drawback in such cases is the very limited number of observations, which cannot deal with autocorrelation effects (seasonal variation), unevenly spaced sampling intervals, or the mathematical complexity of evaluating a vast number of metabolites according to their longitudinal variation. 

Currently, two approaches have arisen in the literature for treating time-dependent metabolomic data. For the first approach, several algorithms have been proposed to deal with the issue based on the data treatment via the incorporation of smoothing functions (i.e., cubic splines, polynomials, LOESS). Two software packages, namely, Time-omics [20] and MetaboClust [21], are available for the abovementioned approach. The second approach involves ANOVA-based algorithms such as the ANOVA–Simultaneous Component Analysis (ASCA) methodology [22]. 

The aim of this research was to examine the plausible differentiations between plants that had been irrigated with the novel onion-based fertilizer versus untreated plants (control samples). The examination of the fertilizer’s biochemical effect on the plant will provide an overview of its efficacy in a hypothesis-free experiment. Thus, the connection between the fertilizer application and its effect on plant growth demonstrated that alternative soil treatment products could benefit plants, paving the way for a new field of fertilizer preparation with a focus on a circular economy. According to our best knowledge, this is the first study to monitor the effect of a compost fertilizer on plant growth. Metabolomics, in conjunction with chemometrics, is the appropriate tool for unveiling the chemical spaces produced by plants. Another equally important objective of the current study was the efficient combination of multivariate chemometrics with time trend analysis in order to contribute to this field, which is largely unexplored.

## 2. Results

### 2.1. Chemometric Results 

A series of five time points was utilized for the investigation of the induced alterations on the plants’ metabolome. The time point t0, which was the initial point of the sampling, was set on the 38th day after the planting of the seeds. At this time point, the plants have four to five leaves. The time points t15, t30, t45, and t60 correspond to 15 days, 30 days, 45 days, and 60 days, respectively, after the starting time point (t0). The leaves of tomato plants that were irrigated with the onion-based fertilizer will be referred to as “onion samples” in this manuscript. On the other hand, the leaves of the tomato plants that were not been treated with the fertilizer will be coded as “control samples”. Six replicates were collected for each matrix (control and onion samples) at each time point.

#### 2.1.1. Multivariate Analysis of Data Obtained by +ESI

Multivariate statistical analysis was deemed necessary for the data analysis due to the complexity of the experimental design. Therefore, Principal Component Analysis (PCA) was employed. The results showed no outliers, but various time trends could be easily noticed, whereas the validity of the analysis was verified as the QC’s formed a tight cluster close to the center of the plot. The locations of the QC samples are illustrated in Appendix A for the control and onion samples in positive and negative ionization mode, respectively. O2PLS-DA was employed to better understand the clustering observed (R^2^Y = 0.532, Q^2^ = 0.391). The model’s validity was assessed using permutation testing and was found to be acceptable, with permutated values of R^2^ and Q^2^ of 0.349 and (−0.207), respectively. The model’s efficiency for proper classification was verified by CV-ANOVA and was evaluated by the corresponding *p*-values (*p*-value = 1.97 × 10^−5^) and the Fisher probability test (Fisher probability practically tends to zero) resulting from the misclassification table, showing no false classifications.

The number of clusters should be assigned correctly to verify that our model is valid. Therefore, an orthogonal unbiased methodology for assigning the proper clustering was employed via the hierarchical analysis of the O2PLS-DA results. As illustrated in Figure 1b, 10 clusters corresponding to the time points from each matrix (control and onion samples) were accurately assigned using this methodology. The pruning towards the formation of the clusters was carried out according to the length of the branching, which revealed the main trends related to plant growth. Hence, the time points t30, t45, and t60 for the onion-irrigated samples are markedly different (Figure 1b). This proves that the effect of the fertilizer starts to be evident after t15. The different time points for the control and onion samples are classified properly, based on the score plots (Figure 1a). Additionally, no statistically significant difference between the aforementioned samples at time point t0 was observed, which was expected, as t0 was sampled before the application of the product.

In order to compare the metabolic differences imposed on the plant, excluding the mutual cross-contributions from the opposite group, O2PLS-DA was performed separately for the onion and control samples, including all of the investigated time points, as shown in Figure 1c,d. At time points t30, t45, and t60, the plants irrigated with the onion-based fertilizer showed no actual differentiation at the first component, whereas t45 and t60 can only be distinguished from t30 at the second principal component. Coalescence at t45 and t60 was observed, indicating that the metabolome does not change significantly at these time points, implying that their growth process is constrained and indicating that plant growth was completed up to the former time point (i.e., t45). A clear discrimination is observed between three time points (t0, t15, and t30), as each group is in a different quartile, a fact that is depicted in Figure 1d. Furthermore, the maximum difference between the time points can be observed between the t0 onion and the t30 onion as the maximum distance occurs, and they can be distinguished at the first and second principal components (Figure 1d).

Regarding the control samples, differentiation can be observed for t15, t30, t45, and t60 at the first principal component, whereas t0 is differentiated from the others at the second principal component. After the 45th day, slow differentiation is still visible, indicating that the metabolism continues to evolve and is consistent with the growth procedure. Furthermore, the time points in the control samples are more tightly clustered than their counterparts in the onion samples. This difference could be attributed to the contribution of fertilizer-induced growth enhancement. The difference between the various time points of the control samples is related to the plants’ normal growth, whereas the difference between the various time points of the onion samples is caused by the additive effect of both their natural growth and the use of the fertilizer (Figure 1c), which may also include an interaction term. 

Pairwise comparison was employed for obtaining interpretable chemometric results, revealing important variables between time points (i.e., t0 vs. t15). The pairwise comparisons (t0 vs. t15, t0 vs. t30, and t15 vs. t30), alongside their corresponding S-plots and their permutation testing, are shown in Figure 2. The Figures of Merit of the classification model for control and onion samples for all time points, as well as pairwise comparisons, are provided in Appendix A. Additionally, the pairwise comparisons for the time points t30 vs. t45, the S-plots, and their corresponding permutation testing in both ionization modes are presented in Appendix A.

#### 2.1.2. Multivariate Analysis of Data Obtained by ESI−

The above-described procedure was followed for the negative ionization mode. Score plots of the control and onion samples (Appendix A), hierarchical clustering analysis for the two groups (Appendix A), O2PLS-DA score plots for the control samples (Appendix A), and O2PLS-DA score plots for the onion samples (Appendix A) using UV scaling at five timepoints (t0, t15, t30, t45, t60) are presented in the Appendix A. 

The statistical treatment exhibits that the influence of the fertilizer differentiates the development process of the plant (Appendix A). Additionally, as illustrated in Appendix A for the control samples, t0 is discriminated from the other time points at both the first and second principal components, while t15 is only distinguished from the others at the first component. No obvious difference is seen among the time points t30, t45, and t60. For the onion-irrigated samples, as shown in Appendix A, the time points t0, t15, and t30 are differentiated at both principal components, although t45 and t60 coalesce. Similar trends were observed from the data derived from both ionization modes. However, the number of features mined from the alignment list in ESI+ is notably higher, which presumably leads to models exhibiting higher confidence.

Pairwise O2PLS-DA comparisons were performed for the time points (a) t0 vs. t15 (b) t0 vs. t30 (c) t15 vs. t30; their score plots, as well as their corresponding S-plots, are presented in Appendix A. The validity of the classification models is proven by the Figures of Merit tabulated in Appendix A.

### 2.2. Metabolites Annotation in Both Ionization Modes

The most influential variables derived from the S-plots in both ionization modes were annotated with the aid of the freely available annotation tool Sirius. The metabolites that were identified can be classified to several categories and may have an impact on plant growth. Specifically, three steroidal alkaloids and their glucosides (tomatidine, tomatidine galactoside, and solasodine), one fatty acid (stearic acid), two organic acids (chlorogenic acid, quinic acid), two flavonoids and their metabolites (quercetin, rutin (quercetin rutinoside)), one lipidglycerol (diacyl glycerol 32:2), and one chlorophyl degradation product (epoxypheophorbide a2) were identified.

The trend of these metabolites and their clustering were investigated via MetaboClust (version 1.2.2.0). The procedure described permits compounds with similar time profiles to be grouped together. The compound’s name, chemical formula, experimental retention time, experimental and theoretical *m*/*z* values, and ionization mode are presented in Table 1. The CSI finger ID score, similarity score derived from SIRIUS, identification level of confidence based on Schymanski et al. [23], and clustering as provided by MetaboClust are also tabulated in this table.

The overlaid time-series diagrams of the annotated metabolites for the control (in red) and onion samples (in green) at five time points (t0, t15, t30, t45, t60) in both ionization modes are depicted in Figure 3. The cluster inclusion of the annotated compounds in ESI+ are illustrated in Appendix A, while the corresponding ones for the annotated compounds in ESI- are depicted in Appendix A. 

## 3. Discussion

### 3.1. Optimization of Sample Preparation

Metabolomic experiments should be focused on detecting as many metabolites in a sample as possible. In this direction, the maximum number of features acquired was used as a criterion to select the most efficient extraction. To decipher the influence of the extraction solvent, the same sample preparation was applied differing only for this parameter. The solvents examined were MeOH, H_2_O:MeOH, 70:30 (*v*/*v*), and Chloroform: MeOH:H_2_O, 20:60:20 (*v*/*v*/*v*). The samples were homogenized and lyophilized. The procedure employed was as follows: Fifty milligrams of the lyophilized samples were weighed, and 1 mL of the abovementioned solvents was added to the samples in each case. The samples were vortexed vigorously for 1 min and then placed in an ultrasonication bath for 15 min at room temperature. Following a 5 min centrifugation at 4000 rpm, the supernatants were collected and filtered through RC syringe filters. The extracts were further diluted (10-fold) with ultra-pure water and transferred to 2 mL autosampler glass vials. The extracts (5 μL) were injected into the UPLC-QToF-MS in both ionization modes. The raw data were calibrated via the Data Analysis software (version 6.0) (Bruker Daltonics, Bremen, Germany), converted by the ABF converter to the abf format [24], and then uploaded to the open-source MS-DIAL software (version 4.9.221218) [25,26], which was used for the estimation of the number of features in each case. The number of features in the positive ionization mode was 804, 1036, 1061 for MeOH, H_2_O:MeOH in a 70:30 (*v*/*v*) ratio, and Chloroform: MeOH:H_2_O 20:60:20, *(v*/*v*/*v*), respectively, whereas in the negative ionization mode, the number of features was 71, 145, and 130 for MeOH, H_2_O:MeOH in a 70:30 (*v*/*v*) ratio, and Chloroform: MeOH:H_2_O 20:60:20, (*v*/*v*/*v*), respectively. Taking into consideration the abovementioned process, H_2_O:MeOH in a proportion of 70:30 (*v*/*v*) was selected as the most efficient extraction solvent.

A second experiment was conducted to investigate the effects of lyophilization, shaking, and sonication. The optimum extraction solvent H_2_O:MeOH, 70:30 (*v*/*v*) was used for the extraction. The features obtained for each sample preparation were 898, 984, and 1173 in ESI+ for the lyophilized samples that had undergone sonication (30 min), for the non-lyophilized samples that were sonicated (30 min), and for the non-lyophilized samples that were only shaken (45 min), respectively. Furthermore, the number of features was 528, 566, and 555 for the sonicated lyophilized, sonicated non-lyophilized, and shaken non-lyophilized samples, respectively, in ESI-. The moisture percentage of the leaves (77 ± 1.9%) was taken into consideration to adjust the initial weight of the fresh samples. Taking into consideration the aforementioned results, it can be assumed that the removal of the solvent during lyophilization may have diminished the number of features that could be detected, since substances may interact irreversibly in the solid phase. This either causes the formation of insoluble or difficult-to-detect complexes. Additionally, based on the obtained results, it is evident that the sonication procedure affects the process negatively, as it leads to a lower number of features. This effect could presumably occur due to the dissipation of excess energy to the molecules, which leads to their thermal decomposition. It could be recommended, at least for such samples, that sonication or use under controlled conditions, i.e., using an ice bath, should be avoided.

### 3.2. Chemometric Strategy

The chemometric approach of the time-series studies was complex. Performing all of the pairwise comparisons (i.e., between the time points of the control and onion samples) may be ineffective and could lead to an enormous comparison data matrix that is difficult to interpret. Therefore, a reduced approach was selected with the aim of providing meaningful results. Thus, the maximum growth data point was located from the O2PLS-DA score plot diagrams (i.e., the time point with the largest distance from the preceding and following one derived from pairwise comparisons). These comparisons were performed for the fertilizer-treated samples. The S-plots from these comparisons showed the metabolites with the largest deviations. We attributed this to the fact that these metabolites were responsible for the developmental evolution of the tomato plants. In order to incorporate the participation of the control samples, trend analysis was performed for all data points (t0, t15, t30, t45, and t60). Finally, the trend analysis results were interrogated for their statistically significant importance for the variable highlighted in the previous procedure.

### 3.3. Role of the Identified Compounds in Plant Health and Their Trends

The primary goal of the current study was to demonstrate how fertilizer irrigation resulted in differentiating plant growth. A metabolomic approach was employed in an attempt to comprehend the mechanism underlying the growth, which may have demonstrated the metabolites that might be expressed differently in the control and onion groups. This is a complementary approach to the assessment of plants’ phenotype characterization. Generally, the trend is focused on the time point t30, as highlighted by the O2PLS-DA. However, some differentiation was noticed for the other time points using MetaboClust. Finally, the time points that were statistically differentiated were examined.

A series of substances were highlighted as markers of growth. Tomatidine acts as a defensive agent, protecting the plant from insects, bacteria, parasites, viruses, and fungi [27]. In accordance with scientific research, tomatidine levels decrease during plant growth as they are converted to their glucoside analogs [28,29]. Based on the diagram in Figure 3a, the systematic decrease in substance levels demonstrates that the plant is developing faster as a result of the fertilizer application. At time points t45 and t60, they are differentiated statistically at 90 percent confidence level, with obtained *p*-values of 0.066 and 0.105, respectively.

The galactoside of tomatidine follows the opposite trend of the corresponding aglucone, as it is found to be at higher levels in the irrigated fertilizer plant (Figure 3b). Tomatidine galactoside is produced by the activity of the GlycoAlkaloid MEtabolism 1 (GAME1) glycosyltransferase [30]. This result verifies the observation that the novel fertilizer enhances the growth rate of the tomato plant compared to the controls, as it demonstrates faster anabolism of the substance compared to the control. 

Solasodine decreased in an analogous matter similar to tomatidine. Based on the biosynthesis pathway “biosynthesis of alkaloids derived from terpenoid and polyketides of KEGG” (accessed on 13 July 2023) (map010666), the *Solanum* alkaloids pathway map shows that tomatidine belongs to the same route as solasodine, only being differentiated in one unsaturation to the b-ring of the steroidal skeleton. Therefore, the same trend in the time series is followed, which might partly justify the conclusion that the onion-irrigated plants had faster growth compared to the untreated ones (Figure 3c).

According to the obtained trend diagram, it appears that at time point t30, which was highlighted from O2PLS-DA as the point of maximum plant growth acceleration, the quercetin levels showed a statistically significant difference (increase) compared to the control ones (Figure 3d). Quercetin promotes plant growth [31]: it was found that spraying quercetin-3-O-rutinoside (rutin) boosts plant development and fruit production, while the antioxidant effects of the substance improve the plants’ health and agility [32,33,34]. It is also a general remark that the antioxidant capacity is tightly linked to the enhancement of plant growth [35]. Finally, it was found that flavonoids, such as quercetin, may act as attractants, contributing to the germination of the plants [36].

The 16:3 moiety of lipids is quite widespread in the *Solanaceae* family [37]. Lipids are used in mitochondrial evolution pathways for energy storage and mobilization. An interesting trend with a quite significant increase in onion samples was observed for the compound diacyl glycerol 32:2 at time point t30. This indicates the mobilization of the lipids, presumably in order to cover the increased energy demands of the plant, which can be correlated with the fast growth of the investigated organism. It is noteworthy that this trend was maintained until the time point t60, which was defined as the end of the experiment. The above-described results are illustrated in Figure 3e. 

Epoxy pheophorbide, a colorless product of chlorophyll degradation connected to the loss of green color, followed a declining trend from the initial time point (t0) of the plants’ irrigation with the fertilizer and throughout the experiment (t60), as illustrated in Figure 3f. Based on the chlorophyll pathway, we come to the conclusion that the abovementioned degradation product is diminished by the application of the novel fertilizer. This may indicate the slower catabolism of chlorophyll and the contribution of fertilizer in improving plant health [38].

Chlorogenic acid, a well-known antioxidant, is vital for the chemical defense against insect herbivores [39]. Furthermore, a plant’s antioxidant capacity is crucial during the early stages of development. Moreover, there are some indications that chlorogenic acid’s conversion to caffeic acid may produce lignans [40], which are essential for the formation of the cell wall. Chlorogenic acid levels only increased at t30 and declined to levels below that of the control at the other time points, according to the trends shown for this compound. The only data that show significant differences between the two groups is t30 (i.e., at t30, the concentration of chlorogenic acid was higher in the onion samples than in the control ones), as illustrated in Figure 3g. The trend analysis performed by MetaboClust revealed that this trend may be significant, since it may be linked to the enhancement of plant growth.

The use of rutin has been proposed as a means of enhancing tomato plant growth, as it promotes the photosynthetic procedure as well as the levels of vital primary plant metabolites such as chlorophyll, carbohydrates, and protein content [32]. Rutin is also a powerful antioxidant, which, as mentioned before, enhances the plant growth [41]. According to the diagram in Figure 3h, a transient, statistically significant increase was observed for t15 and t30 for rutin, whereas a consistent decrease was noticed till the end of the experiment. Statistically significant important differences were observed for the time points t15 (*p*-value = 0.0015, confidence level 95%) and t30 (*p*-value = 0.10, confidence level 90%) between the control and the onion samples. Furthermore, at t60, statistically significantly lower levels (*p*-value = 0.021, confidence level 95%) were noticed for the onion samples. 

Quinic acid exhibits antioxidant and synergistic activity in conjunction with numerous compounds such as quercetin [42] and undecanoic acid [43]. Furthermore, quinic acid is shown to chelate metals, which is in accordance with previous observations [44]. There are literature references correlating the growth and the levels of quinic acid in kiwi fruit, supporting that fruit size is increased with increasing substance levels, eventually reaching a plateau [45]. Therefore, quinic acid is considered to play a significant role in plant growth. According to Figure 3i, a statistically significant difference was noticed between t15 and t30 (*p*-value = 0.011) for the onion-irrigated plants, whereas for the rest of the timepoints, there appeared to be a plateau, indicating the completion of the growth. On the other hand, for the control samples, the peak was reached at the t45 time point, demonstrating a slower rate of growth. 

Fatty acids are essential components of cellular metabolism, as they are used for the production of energy via beta-oxidation. Furthermore, they are used as building blocks of the cellular membrane [46]. According to the diagram in Figure 3j, stearic acid was found to decrease in the onion samples while remaining constant in the control samples. Statistically, significant differences were observed for the time points t30, t45, and t60 (*p*-value = 6.92 × 10^−6^, 0.0018, and 2.64 × 10^−6^, respectively) between the investigated groups (i.e., the control samples vs. the onion samples at the respective time points). This indicates that the onion-irrigated plants, which showed lower levels of the acid, used stearic acid intensively. This is in accordance with the use of the molecule as fuel for the increased energy demands of the process, as well as its consumption for constructing the cellular membranes of newly formed cells.

## 4. Materials and Methods

### 4.1. Leaf Harvesting and Time Point Definition

The tomato seeds were planted using supplied soil with perlite in proportion (4:1) in a greenhouse. The temperature was set at 25 ± 2 °C and the plants were watered on a daily basis using an automatic watering mechanism. Based on the life cycle of the tomato plant, five time points (t0, t15, t30, t45, and t60) were selected for plant health/growth monitoring. The application of the fertilizer was performed without any further processing of the product. The procedure for obtaining the onion-based fertilizer was described in detail in the authors’ previous research [3]. Control samples were used at each time point to eliminate cofounding parameters due to the plant’s normal growth. The initial time point for leaf ripening (t0) was defined 38 days after plant cultivation, when a sufficient quantity of leaves was available for the experiment to proceed. The time points t15, t30, t45, and t60 correspond to 15 days, 30 days, 45 days, and 60 days, respectively, after the starting time point (t0).

Special care was given to the termination of metabolomic functions in plants. The leaves were gently harvested, immediately placed in liquid nitrogen, and then stored at −80 °C until analysis, according to literature recommendations [47]. At each time point, leaves from three different heights of the plant (low, medium, and high) were harvested, pooled, and place in alumina foils. The harvesting of the leaves took place before the daily watering. 

### 4.2. Reagents and Materials

All standards and reagents used were of analytical grade purity (<95%), unless differently stated. Ammonium acetate and ammonium formate were purchased from Fisher Scientific (Geel, Belgium). Methanol (MeOH) (LC–MS grade) was obtained from Merck (Darmstadt, Germany), while formic acid 99% and acetic acid were provided by Fluka (Buchs, Switzerland). The ultrapure water (H_2_O) was provided by a Milli-Q device (Millipore Direct-Q UV, Bedford, MA, USA). Regenerated cellulose syringe filters (RC filters, pore size 0.2 μm, diameter 15 mm) were acquired from Macherey-Nagel (Düren, Germany). Yohimbine, reserpine, and 4-aminosalicylic acid were purchased from Sigma-Aldrich (Stenheim, Germany). Stock solutions of the reference standards (1g L^−1^) were prepared in MeOH (LC-MS grade) and stored at −20 °C in ambient glass containers. Working solutions at a concentration of 40 mg L^−1^ were prepared by appropriate dilutions of the stock solutions with MeOH.

### 4.3. Sample Preparation for HRMS Analysis

The leaves were homogenized, and 0.25 g were weighted into 15 mL centrifuge tubes. A volume of 3.5 mL MeOH, including the internal standards (resperidine, yohimbine, and 4-amino salicylic acid at a concentration of 12.0 mg L^−1^) was added to the samples. The samples were vortexed for 30 s and the addition of 6.5 mL ultrapure water followed. Consequently, they were vortexed for 30 s and shaken for 30 min on a horizontal shaker. The samples were centrifuged at 4000 rpm for 5 min, and the supernatants were filtered through RC syringe filters. The extracts were diluted two-fold with the addition of a H_2_O: MeOH, 70:30 (*v*/*v*) mixture and 5 μL of the extract was injected into UPLC-QToF-MS in both ionization modes.

### 4.4. Instrumental Analysis

The analysis for the elucidation of the differentiated metabolites in the plant leaves was carried out using an ultra-high-performance liquid chromatograph (UHPLC) equipped with an HPG-3400 pump (Dionex Ultimate 3000 RSLC, Thermo Fisher Scientific, Dreieich, Germany) coupled to a time-of-flight mass analyzer (Hybrid Quadrupole Time of Flight Matic Bruker Daltonics, Bremen, Germany). An Acclaim RSLC 120 C18 column (2.2 μm, 2.1 × 100 mm, Thermo Fisher Scientific, Dreieich, Germany) was used, equipped with a pre-column (Van guard Acquity UPLC BEH C18 (1.7 μm, 2.1 × 5 mm, Waters, Ireland). The column temperature was set at 30 °C throughout the chromatographic analysis, while the injection volume was 5 μL. The analysis was performed in both ESI modes (positive and negative). The mobile phases in the positive ionization mode consisted of: (A) aq. 5 mM ammonium formate: MeOH (90:10 *v*/*v*) acidified with 0.01% formic acid, and (B) 5 mM ammonium formate in MeOH acidified with 0.01% formic acid, and, in negative ionization mode, were (A) aq. 10 mM ammonium acetate: MeOH (90:10 *v*/*v*) and (B) 10 mM ammonium acetate in MeOH. Τhe LC and MS settings are described in detail in previous work by our group [3]. Both data-independent (broad-band Collision Induced Dissociation -bbCID: DIA) and data-dependent acquisition (AutoMS, DDA) modes were utilized. The MS and MS/MS spectra were obtained using two different collision energies (4 eV and 25 eV) in the bbCID mode. In AutoMS mode, the five most abundant ions per MS scan were selected and fragmented using ramp collision energy.

Quality control (QC) samples were prepared and analyzed at the beginning as well as during the sequence in both acquisition modes. The QC samples were analyzed at the beginning of the sequence as an indicator of instrument stability concerning the retention time and the signal intensity. Additionally, QC samples were used during the sequence for batch correction due to the drift of the instrument. Two QCs were prepared related to the matrix investigated, receiving equal quantities of each matrix at each time point. Therefore, one QC sample for the control samples and one QC sample for the leaves irrigated with the onion-based fertilizer were prepared. Additionally, one procedure blank at each time point was prepared.

### 4.5. Mass Spectrometry Data Analysis

The workflow developed for the current study is depicted in Figure 4. Each step is analyzed below.

#### 4.5.1. Screening Methodology

The raw data were calibrated via Data Analysis software (Bruker Daltonics, Bremen, Germany), converted using the ABF converter [24], and then uploaded to the open-source MS-DIAL software (version 4.9.221218) [25,26]. The data derived from data-dependent (autoΜS) and data-independent (bbCID) acquisition modes were processed separately. The sequence of the samples was noted, as normalization was conducted with QC samples and internal standards (LOWESS and internal standard normalization). The exported list (peak area, retention time, and *m/z*) of features was imported to SIMCA 14.1 for chemometric analysis. 

#### 4.5.2. Chemometrics Methodology

The annotation list was derived from MS-DIAL, and the information (*m*/*z*, retention time, peak area, class) was used to build the SIMCA dataset. PCA hierarchical analysis and O2PLS-DA were carried out, with various scaling parameters (UV, pareto, no scaling) employed. The predictive value (Q^2^) was calculated using seven-fold cross validation, while R^2^ was estimated using all of the observations. The model’s credibility was assessed via Figures of Merit (R^2^, Q^2^, CV-ANOVA, *p*-value, permutation testing for 100 random permutations, and confusion matrix). O2PLS-DA was performed using UV scaling on the control and onion samples (t0, t15, t30, t45, t60), and pairwise O2PLS-DA was performed for the onion samples (t0 vs. t15, t15 vs. t30, and t0 vs. t30) using pareto scaling. Their corresponding S-plots were constructed to elucidate the model’s most highly contributing features. To investigate the trend of these features, MetaboClust was used.

#### 4.5.3. Time Series

MetaboClust is an open-source software program used to analyze time-course metabolomics [21]. The peak intensities, observations, and the peak information table, that is, the data inputs for the software, were organized in csv files. The peak-picking process, which was carried out using the MS-DIAL, generated an msp file including information such as the peak areas per sample for the associated feature (*m*/*z* and retention time). These pieces of information were included in the csv file called “IntensityData” based on the requirement of the software. Furthermore, the “Observation Info” csv file comprised experimental information such as sample names, the experimental group (e.g., control, onion), the time point (e.g., t0, t15, t30, t45, t60), the number of replicates per group and time point, as well as the batch and acquisition order. The “Peak Info” csv file, which contained information regarding *m*/*z*, retention time, and the ionization mode, was also required. Finally, the annotation file contained the name of the metabolite alongside its related *m*/*z* and retention time.

Batch correction was accomplished using UV scaling and centering, while trend fitting was realized using LOESS, where the span was set to 5. Metabolite clustering was achieved using the k-means Hartigan Wong algorithm based on Euclidean distance, where the number of clusters was defined to 10 (k = 10).

#### 4.5.4. Metabolite Annotation

Sirius was utilized for metabolite annotation [48]. The list of the features was exported from MS-DIAL (msp format) and imported to Sirius 5.8.1. Briefly, the identification was based on the accurate mass and the MS/MS spectra. Thus, MS-DIAL software (version 4.9.221218) was used to construct the extracted ion chromatograms, but mainly to deconvolute the MS/MS spectra, which were then subjected to the Sirius 5.8.1-based analysis. Various databases were implemented, such as PlantCyc, BioCyc, Coconut, GNPS, PubChem, Natural Products, KNApSacK, and KEGG, for molecular formula and structure identification. CANOPUS was used for predicting the compound class using the MS/MS spectra [49]. CSI:FingerID [50] and the similarity score implemented were used for the evaluation of the annotated metabolites. The identification level of confidence was ranked based on Schymanski et al.’s scheme.

## 5. Conclusions

A comprehensively characterized onion-based fertilizer was utilized for the irrigation of tomato plants to investigate its effect on plant growth. Differentiation between the control and the onion-irrigated plants was highlighted via the investigation of a five-time-point experiment based on the tomato cycle life. The identified compounds, belonging to various categories such as steroidal alkaloids and their glucosides, organic acids, fatty acids, flavonoids, and their metabolites, act beneficially for plant growth. These metabolites were derived by applying a newly developed workflow combining multivariate chemometrics (O2PLS-DA) and time trend analysis of the most important variables. In order to facilitate of the adoption of the proposed workflow from the scientific community, open-source software was employed for peak picking (MS-DIAL), annotation (SIRIUS), and time-series monitoring (MetaboClust). Employing advanced analytical methods (i.e., HRMS-based metabolomics) provides the opportunity to gain holistic information about plant development, as these approaches reveal how the underlying biochemistry is affected by the corresponding intervention. This kind of information acts in a complementary and synergistic way alongside the established methods of agricultural experimentation, such as phenotype monitoring.

The results indicate that the presence of the identified compounds could ameliorate plant health, paving the way for the holistic monitoring of plant growth. Finally, this research highlights the significance of developing by-product-based fertilizers, exploiting otherwise neglected materials and ultimately contributing to the cyclic economy.

## Figures and Tables

**Figure 1 molecules-28-06771-f001:**
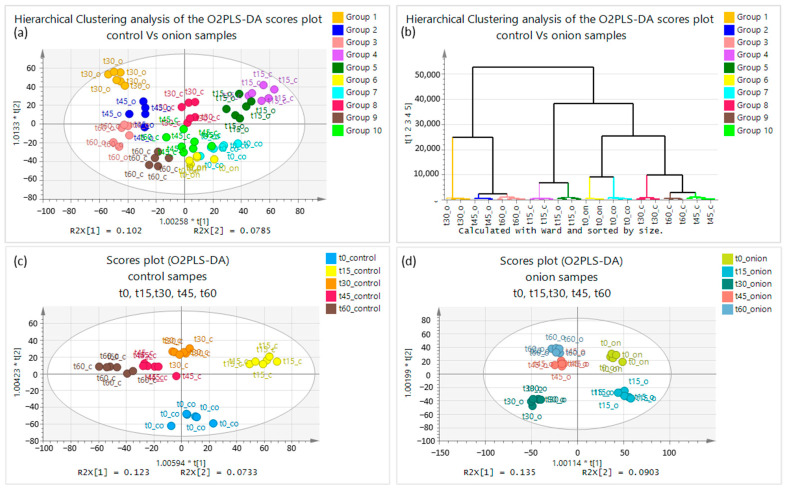
Plots of O2PLS-DA and hierarchical clustering analysis: (**a**) score plot of control and onion samples; (**b**) hierarchical clustering analysis O2PLS-DA of control and onion samples; (**c**) score plot O2PLS-DA discrimination using UV scaling between control samples; (**d**) score plot O2PLS-DA discrimination using UV scaling for onion samples at five time points (t0, t15, t30, t45, t60) in positive ionization mode.

**Figure 2 molecules-28-06771-f002:**
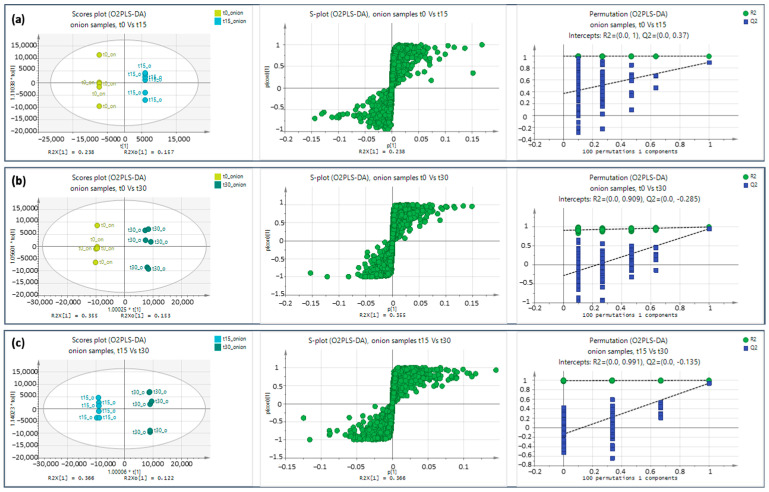
Score plot O2PLS-DA discrimination using pareto scaling, S-Plot, and permutation testing for onion samples at (**a**) t0, t15, (**b**) t0, t30, (**c**) t15, and t30 in positive ionization mode.

**Figure 3 molecules-28-06771-f003:**
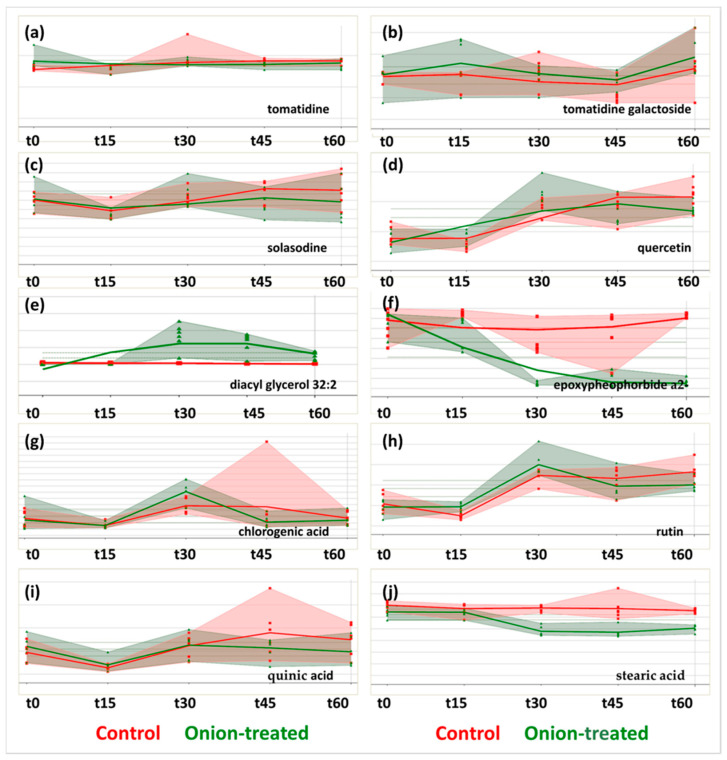
Overlaid time-series diagrams of control (in red) and onion samples (in green) using MetaboClust for (**a**) tomatidine, (**b**) tomatidine galactoside, (**c**) solasodine, (**d**) quercetin, (**e**) diacyl glycerol 32:2, (**f**) epoxypheophorbide a2-, (**g**) chlorogenic acid, (**h**) rutin, (**i**) quinic acid, and (**j**) stearic acid at five time points (t0, t15, t30, t45, t60). The bold red and green lines are the result of using the LOESS smoothing algorithm (span = 5). The red and green dots correspond to the replicates (*n* = 6) at each time point. The red and green shaded areas correspond to the convex hulls of the respective control and onion samples.

**Figure 4 molecules-28-06771-f004:**
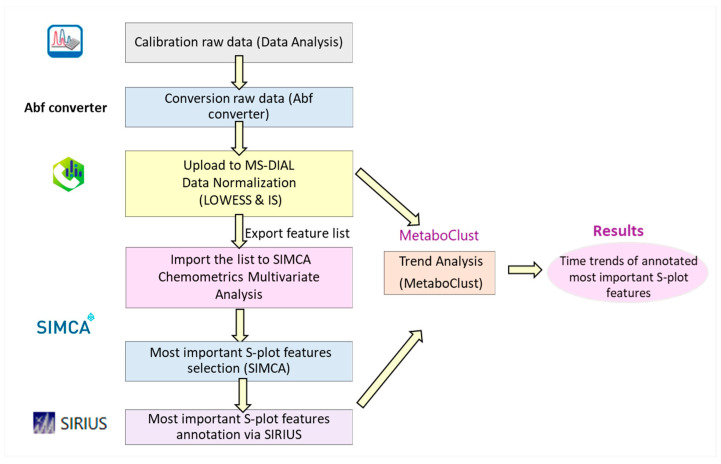
Workflow of the developed methodology for the identification of the most important metabolites for plant growth.

**Table 1 molecules-28-06771-t001:** Most influential variables identified via non-target screening.

Compound Name	Chemical Formula	Exp. t_R_ (min)	Exp. *m*/*z* Values	Theor. *m*/*z*	ESI Mode	CSI Finger ID Score	Similarity	Level ^a^	Cluster k
tomatidine	C_27_H_45_NO_2_	8.0	416.3548	416.3523	+ESI	−110.14	59.8	2a	3
tomatidine galactoside	C_33_H_55_NO_7_	7.3	578.4055	578.4051	+ESI	−94.64	67.8	2a	6
solasodine	C_27_H_43_NO_2_	7.7	414.3384	414.3367	+ESI	−150.3	54.1	2a	2
quercetin	C_15_H_10_O_7_	6.8	303.0506	303.0499	+ESI	−31.2	84.3	1	2
diacyl glycerol 32:2	C_35_H_66_O_4_	15.6	551.5031	551.5034	+ESI	Lipid map		3	3
epoxypheophorbide a(2−)	C_35_H_34_N_4_O_6_	14.0	607.2529	607.2551	+ESI	TomatoCyc		3	7
quinic acid	C_7_H_12_O_6_	1.3	191.0567	191.0561	−ESI	−67.6	47.7	1	5
stearic acid	C_18_H_36_O_2_	14.5	283.2654	283.2643	−ESI	−11.3	100	2a	3
chlorogenic acid	C_16_H_18_O_9_	2.9	353.0873	353.0878	−ESI	−8.1	100	1	1
rutin	C_27_H_30_O_16_	5.7	609.1459	609.1461	−ESI	−57.98	89.2	1	9

^a^ Level of identification confidence based on Schymanski et al. [23].

## Data Availability

Data are available upon request.

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
