# Peer review of "Longitudinal Plant Health Monitoring via High-Resolution Mass Spectrometry Screening Workflows: Application to a Fertilizer Mediated Tomato Growth Experiment"

_molecules, 2023, doi:10.3390/molecules28196771_

Round 1

Reviewer 1 Report

The article written by A. Panara et al. is based on the use of HRMS and statistical tools study the effect of an onion-based fertilizer on tomato plant growth. The title is clear, discussion is of real interest, language is correct, and the topic is very attractive. Nevertheless, it needs some minor improvements before it can be published in Molecules.

In detail, I have the following comments:

The abstract and introduction are clear, introduction presents the subject and interest of this work.

The results section is well written but the unit of time used for this experiment is missing throughout the manuscript. it is mentioned only once, on line 137 (“45th day”) but never before or after. Please add this information at the beginning.

The gap in the time point range seems to be too wide as in positive mode, an effect observed immediately at the 2nd point of the time line (line 116 “fertilizer starts to be evident after t15”) it would have been good to confirm it with further points at t5 and t10.

Figures use colors that are too close together, making them difficult to interpret (e.g. fig 1a). It would have been preferable to use colors and shapes to differentiate points more clearly.

Figure 2 would have been more complete with a comparison between t30 and t45 to see that the effect over a longer time span.

Results of negative mode are not as clear as those of positive mode. It would have been good to merge positive and negative modes for this type of statistical interpretation.

Line 185 please explain the acronym VIPS.

In section 2.2, compounds were identified on the basis of mass spectrometry results, please add the confidence level (see “Identifying Small Molecules via High Resolution Mass Spectrometry: Communicating Confidence”, Environ. Sci. Technol. 2014, 48, 2097−2098 dx.doi.org/10.1021/es5002105)

The discussion part is very interesting, but it would have been good to add a comment on the effect of lyophilization.

In line 258 there is a double “the” (“distance from the the”) please delete one.

The interpretation of figure 3 with O2PLS-DA is made with a very high p-value (90% of confidence level) please explain why this is not the usual confidence level (95% or even 99%)? The difference doesn’t seem real given the variability of the samples and the very low number of replicates.

The material and methods section is well written and interesting, but lacks the units of the sampling points and the preparation of onion fertilizer (i.e. preparation, concentration, etc.).

Conclusion is too short and could be developed further.

To conclude, the subject is very interesting and promising, the authors discuss all the results in detail. For all the reasons cited above, this article should undergo minor revision before being published in Molecules.

Author Response

Please find attached the responses to the comments.

Reviewer 2 Report

This work presents a HRMS screening workflows for a fertilizer mediated tomato growth experiment. The results and approach are interesting especially monitoring the effect of a compost-fertilizer on plant growth. However, there are some suggestions as follows:

1 the data of QC samples was not displayed in the O2PLS-DA figure.

2 how to identify the compound in table 1.

3 p-value changed to p-value

4 line 359, p-value = 6.92 x 10-6, 0.0018, and 2.64 x 10-6?

Author Response

(The authors gave the same response as above.)

Reviewer 3 Report

I think this paper should be published but it requires some corrections. My comments are in the attached file

·         In general, the entire text should be reviewed to homogenize the tense of the verbs. The separation of words in the line break should also be reviewed.

Author Response

(The authors gave the same response as above.)
